# IL-27 limits HSPC differentiation during infection and protects from stem cell exhaustion

**Daniel L Aldridge[1], Zachary Lanzar[1], Anthony T Phan[1], David A Christian[1], Ryan D Pardy[1], Booki Min[2], Ross Kedl[3], Christopher A Hunter[1]\***

[1]University of Pennsylvania School of Veterinary Medicine, Philadelphia, United States; [2]Department of Microbiology and Immunology, Feinberg School of Medicine, Northwestern University, Chicago, United States; [3]University of Colorado, Anschutz Medical Campus, Aurora, United States

## eLife Assessment

The article presents **important** findings describing the role of IL27 in maintaining HSCs at steady state, and in emergency haematopoiesis in response to T. goodii by limiting the inflammatory monocyte outcomes. The evidence provided are **solid** and support that IL27 acts at the level of HSCs and not downstream. This study will be of interest to immunologists and hematologists, as well as infectious disease researchers.

**\*For correspondence:**
chunter@vet.upenn.edu

**Competing interest:** The authors declare that no competing interests exist.

**Abstract** Many inflammatory stimuli can induce progenitor cells in the bone marrow to produce increased numbers of myeloid cells as part of the process of emergency myelopoiesis. These events are associated with trained immunity and have long-term impacts on hematopoietic stem and progenitor cell (HSPC) development but can also compromise their function. While many cytokines support emergency myelopoiesis, less is known about the mechanisms that temper these events. When mice that lack the cytokine IL-27 were infected with *Toxoplasma gondii*, there was enhanced generation of monocyte progenitors and increased numbers of inflammatory monocytes. In the bone marrow of infected mice, there was increased production of IL-27 that localized with HSPCs, and a survey of cytokine receptor expression highlighted that HSPCs were uniquely poised to respond to IL-27. Furthermore, the use of in vitro differentiation assays and mixed bone marrow chimeras revealed that HSPCs from IL-27-deficient mice are predisposed toward the monocyte lineage. Additional studies highlighted that after infection, loss of the IL-27R resulted in reduced HSPC fitness that manifested as reduced proliferative responses and a decreased ability to reconstitute the hematopoietic system. Thus, the ability of IL-27 to act on HSPC provides a regulatory brake on differentiation to limit monocyte induction and preserve HSPC stemness.

## Introduction

In response to inflammatory stimuli, cytokine production can directly affect progenitor populations in the bone marrow leading to alterations in hematopoiesis (*Collins et al., 2021*). For example, in response to microbial challenges, elevated levels of GM-CSF, IFN-γ, and IL-6 modulate hematopoietic stem and progenitor cell (HSPC), which contain hematopoietic stem cells (HSCs) as well as multipotent progenitors (MPPs), differentiation and enhance myelopoiesis (*Baldridge et al., 2010*; *Kaufmann et al., 2018*; *Khan et al., 2020*; *Link, 2000*; *MacNamara et al., 2011a*; *MacNamara et al., 2011b*; *Maeda et al., 2005*; *Matatall et al., 2014*; *Xie et al., 2021*; *Mirantes et al., 2014*; *Reynaud et al.,*

*2011*). These events can also lead to long-term epigenetic and metabolic changes in HSCs that form the basis for the process of trained immunity (*Ochando et al., 2023*). Thus, many of the cytokines produced during different infections are associated with trained immunity (*Kaufmann et al., 2018*; *Khan et al., 2020*). For example, after challenge with *Mycobacterium avium* or SARS-CoV-2, IFN-γ and IL-6, respectively, can skew HSCs toward myeloid output and increase the functionality of their downstream progeny (*Cheong et al., 2023*; *Matatall et al., 2014*).

While trained immunity is associated with long-term changes to HSCs, inflammation can also reduce HSC functionality, which can manifest as decreased proliferative and differentiation potential. To mitigate these adverse effects, there are cell-intrinsic mechanisms such as autophagic processes to limit oxidative damage, epigenetic reprogramming, and metabolic changes that contribute to the maintenance of stemness (*Ibneeva et al., 2024*; *Liu et al., 2020*; *Ma et al., 2020*; *Revuelta and Matheu, 2017*; *Singh et al., 2018*; *Zhao and Deininger, 2023*). Without these protective mechanisms, HSCs accumulate cellular damage and lose their ability to self-renew and differentiate, a process termed stem cell exhaustion. While inflammation is linked to HSC exhaustion, less is known about the cell extrinsic factors that contribute to the maintenance of HSC quiescence and integrity, but several cytokines have been implicated in this process. For example, in response to repeated stimulation with poly I:C, IFN-α can remove actively proliferating HSCs from the developmental pool as a mechanism to prevent the accumulation of damaged HSCs (*Pietras et al., 2014*). In addition, during fetal development, there is evidence that IL-10 limits HSC responsiveness to inflammatory signaling and thereby limits emergency myelopoiesis as a mechanism to preserve HSC functionality (*Collins et al., 2024*).

IL-27 is a member of the type I family of cytokines that utilize JAK–STAT signaling to mediate their biological effects. This family also includes IL-6, and both of these cytokines signal through dimeric receptors that contain the gp130 receptor combined with unique IL-6Rα and IL-27Rα components. Because T cells constitutively express high levels of the IL-27R (*Pflanz et al., 2004*; *Pflanz et al., 2002*), the majority of studies on this cytokine have focused on its impact on these populations. Thus, IL-27 can limit the development of pathological Th1, Th2, and Th17 CD4[+] T cell responses (*Findlay et al., 2010*; *Hamano et al., 2003*; *Villarino et al., 2003*), but several studies have also indicated that the absence of IL-27 results in an elevated population of inflammatory monocytes that contribute to the development of immune pathology (*Aldridge et al., 2024*; *Liu et al., 2021*). The observation that the loss of IL-27 also results in enhanced inflammatory monocyte induction prior to a detectable T cell response suggests that IL-27 may antagonize the innate events that promote these monocyte populations (*Aldridge et al., 2024*). Consistent with this idea, there are additional reports that IL-27 can limit neutrophil and monocyte responses and directly impact myelopoiesis (*Furusawa et al., 2016*; *Liu et al., 2014*; *Liu et al., 2021*; *Seita et al., 2008*; *Sun et al., 2017*; *Wirtz et al., 2006*). However, the observation that monocytes do not express the IL-27R suggests that the ability of IL-27 to modulate this population is indirect.

The studies presented here show that infection with *Toxoplasma gondii* results in increased activity of HPSC and downstream progenitors associated with emergency myelopoiesis; however, in the absence of IL-27, HSPCs display a skewed developmental program that resulted in enhanced monopoiesis. While all HSPCs showed preferential expression of the IL-27R, with long-term hematopoietic stem cells (LTHSCs) and MPP2s exhibiting the highest, the loss of IL-27 during infection resulted in reduced HSPC fitness. Thus, during infection, the ability of IL-27 to act on HSPCs provides a regulatory brake on differentiation to limit monocyte induction and preserve HSPC stemness.

## Results

### IL-27 regulates monopoiesis during toxoplasmosis

Infection with *T. gondii* leads to increased production of monocytes that is enhanced in the absence of IL-27 (*Aldridge et al., 2024*), but it is unclear if this is a result of differentiation from HSPCs or from the more differentiated monocyte progenitors (MPs). Therefore, the *Procr*[creERT2-IRES-tdTomato] mouse line, which expresses Cre in HSPCs (*Gur-Cohen et al., 2015*; *Iwasaki et al., 2010*), was crossed with the Ai6 reporter line to provide a lineage tracing model. In these mice, tdTomato is expressed in *Procr*[+] cells and, upon tamoxifen (TAM) treatment, HSPCs will be tdTomato[+] and zsGreen[+], but their differentiated progeny will only express zsGreen (*Figure 1A*). For these experiments, Procr-Ai6 mice were treated with TAM, infected, and then at 5 dpi the BM and peripheral cells were analyzed.

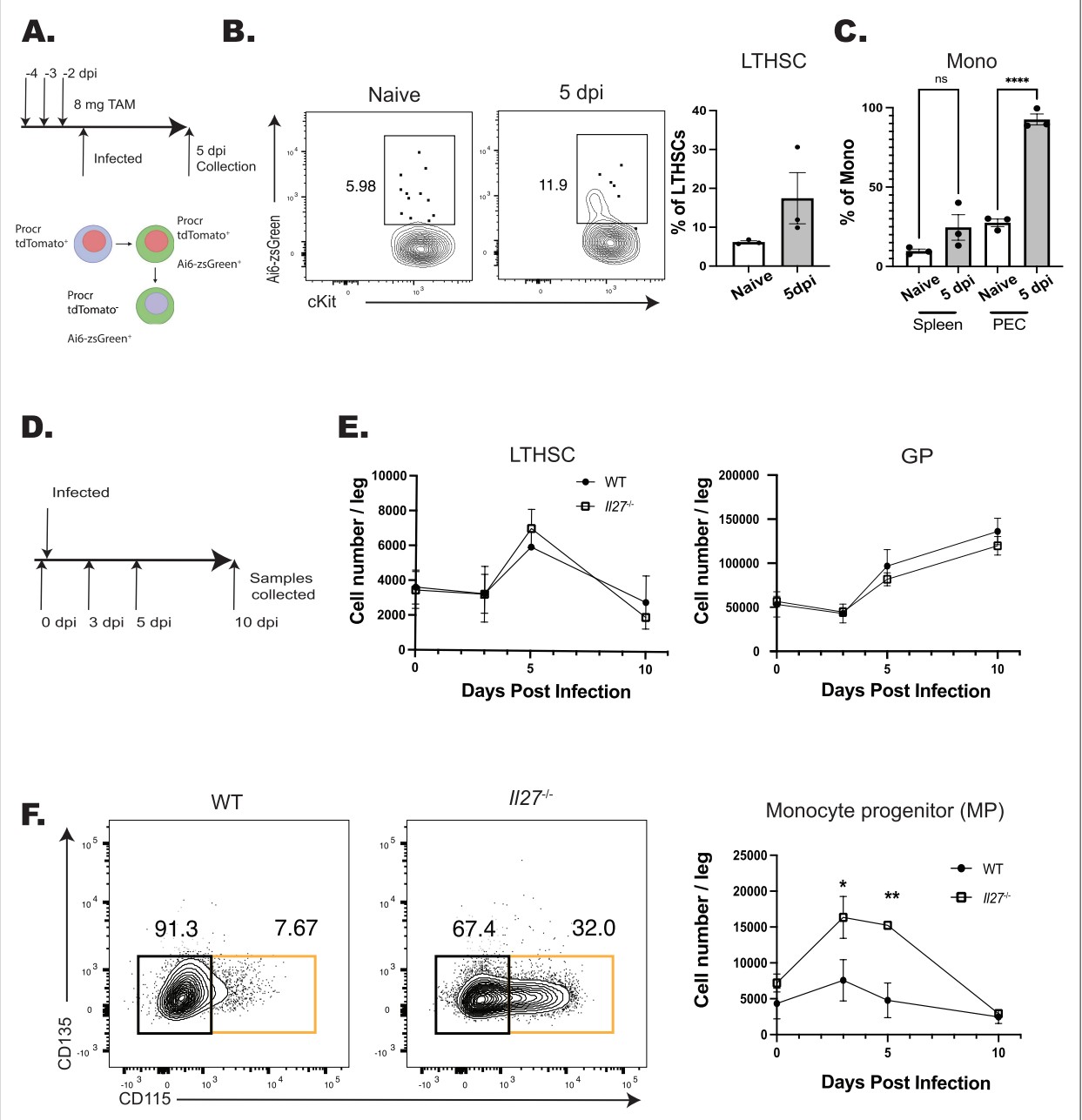

**Figure 1.** IL-27 regulates monopoiesis during infection. (**A**) Schematic of infection and TAM dosing strategy for Procr-Ai6 mice. (**B**) Procr-Ai6 mice were infected and representative flow plots of long-term hematopoietic stem cells (LTHSCs) (Lin (CD3, NK1.1, B220, Ly6G)⁻, Sca-1⁺, CD117 (cKit)⁺, CD135⁻, CD48⁻, CD150⁺) from naive (left) and infected (right) mice are shown. Proportions of zsGreen⁺ LTHSCs were then quantified in both groups. (**C**) Proportions of zsGreen⁺ monocytes (CD3⁻, B220⁻, CD11b⁺, Ly6C⁺, Ly6G⁻) in the spleen and peritoneal exudate cells (PECs) of naive and infected mice. (**D**) Schematic of infection of WT and *Il27⁻/⁻* mice for E and F. (**E**) Numbers of LTHSCs and granulocyte progenitors (GPs) (CD3⁻, NK1.1⁻, B220⁻, CD117⁺, CD34⁺, CD16/32ʰⁱ, Ly6C⁺, CD135⁻, CD115⁻) in the bone marrow of WT and *Il27⁻/⁻* infected mice throughout infection. (**F**) Monocyte progenitors (MPs) (CD3⁻, NK1.1⁻, B220⁻, CD117⁺, CD34⁺, CD16/32ʰⁱ, Ly6C⁺, CD135⁻, CD115⁺; orange box) in the bone marrow of infected WT and *Il27⁻/⁻* mice throughout infection. Representative flow plots are shown (left) and quantified (right). Statistical significance was tested by one-way ANOVA with Sidak's correction. *, **, and *** correspond to p-values ≤0.05, 0.01, and 0.001, respectively. *N* = 3–5 mice/group and data shown are representative of two to three repeated experiments. Error bars indcate standard error of the mean (SEM).

The online version of this article includes the following figure supplement(s) for figure 1:

**Figure supplement 1.** IL-27 regulates monopoiesis during infection.

**Figure supplement 2.** Gating strategy for flow cytometric analysis of cells in the bone marrow and periphery.

Consistent with other HSPC lineage tracing models (*Chapple et al., 2018*; *Säwen et al., 2018*), in naive bone marrow, only a small proportion (~5%) of LTHSCs were labeled, which corresponded to a low proportion of zsGreen+ monocytes in the peritoneum and spleen (*Figure 1B, C*). After infection, the proportion of zsGreen+ LTHSCs in the marrow nearly doubled (*Figure 1B*), and the proportion of monocyte dendritic cell progenitors (MDPs) also increased (*Figure 1—figure supplement 1A*). This corresponded to an increase in labeled monocytes and neutrophils in the bone marrow of infected mice (*Figure 1—figure supplement 1A*). In the periphery, there was also an increased proportion of zsGreen+ monocytes in the spleen, while in the peritoneum, the initial site of infection and monocyte recruitment, the proportion of labeled monocytes reached ~100% (*Figure 1C*). While this proportion of zsGreen+ monocytes is substantial, these data must be interpreted with care as the *Procr*^creERT2-IRES-tdTomato is expressed in multiple progenitor cells—not just HSCs but other MPPs. Thus, these data do not suggest that all recently generated monocytes are derived from HSCs, only that these monocytes come from HSPC differentiation and expansion instead of downstream committed granulocyte monocyte progenitors (GMPs) or MPs. All progenitors and their progeny were identified based on the expression of defining surface markers as measured by flow cytometry (*Figure 1—figure supplement 2*), and additional phenotyping revealed that both classical (CCR2hi and CX3CR1lo) and non-classical (CCR2lo and CX3CR1hi) monocyte populations were labeled to equivalent degrees, indicating that infection impacts the recruitment of both subsets (*Figure 1—figure supplement 1B*). Collectively, these datasets indicate that the increased myelopoiesis that occurs in response to infection with *T. gondii* is due to expansion and enhanced differentiation of HSPCs, and not just MPs, into downstream progeny, eventually resulting in increased monocyte production.

To determine the impact of IL-27 on this infection-induced myelopoiesis, WT and *Il27*−/− mice were challenged with *T. gondii* and numbers of progenitors (from LTHSCs to MPs) were analyzed (*Figure 1D*). While infection led to an expansion of LTHSCs, the numbers of LTHSC, granulocyte progenitors, and other progenitors (including common lymphocyte progenitors, common myeloid progenitors, and GMPs) (data not shown) were not impacted by the absence of IL-27 (*Figure 1E*). Infection also resulted in a transient increase in MPs in WT mice, but in the absence of IL-27, this was enhanced at 3 and 5 dpi before returning to baseline at 10 dpi (*Figure 1F*). A similar pattern was observed for the MDPs (*Figure 1—figure supplement 1C*), and in the *Il27*−/− mice, this corresponded to an increase in mature monocytes in the liver (*Figure 1—figure supplement 1D*). The CCR2hiCX3CR1lo phenotype of these cells indicated that these were inflammatory monocytes that were predominantly within the tissue parenchyma (*Figure 1—figure supplement 1E*), suggesting that these monocytes are poised to participate in the tissue pathology associated with these mice. This enhanced monocyte differentiation appeared independent of IFN-γ, as blockade of IL-27 during infection in IFN-γ reporter mice showed no alteration in cytokine levels in the bone marrow (*Figure 1—figure supplement 1F*). Thus, one function of IL-27 during acute toxoplasmosis is to limit monopoiesis and the generation of inflammatory monocytes.

## IL-27 regulates monocyte development in a cell-intrinsic manner

Previous studies have shown that during toxoplasmosis or trypanosomiasis, the absence of IL-27 results in enhanced CD4+ T cell responses which contribute to increased accumulation of inflammatory monocytes (*Aldridge et al., 2024*; *Liu et al., 2021*). To test if the early (5 dpi) enhanced monopoiesis in the *Il27*−/− mice observed here was a secondary consequence of the CD4+ T cell response, a *CD4*^cre *Il27ra*^fl/fl (CD4-IL-27R) mouse line was generated. In these mice, IL-27R expression was efficiently deleted in T cells, but other IL-27R-expressing cells, such as B cells, maintained expression (*Figure 2—figure supplement 1A*). When infected with *T. gondii*, these mice phenocopied whole body *Il27ra*−/− mice and developed lethal immune pathology >10 days post-infection (data not shown). When control and CD4-IL-27R mice were infected and analyzed at 5 dpi, there were equivalent numbers of MPs (*Figure 2A*). Thus, the enhanced monopoiesis observed in the whole-body *Il27*−/− mice is independent of the effect of IL-27 on CD4+ T cells.

Next, a mixed bone marrow chimera approach was used to determine if the enhanced monopoiesis observed in the absence of IL-27 was cell-intrinsic. In these experiments, WT and *Il27ra*−/− marrow from naive mice was used to generate a series of individual and mixed bone marrow chimeras (*Figure 2B*). In the single-transfer chimeras, at 14 weeks after reconstitution, WT and *Il27ra*−/− marrow reconstituted the hematopoietic system of irradiated recipients with 100% of hematopoietic cells derived from

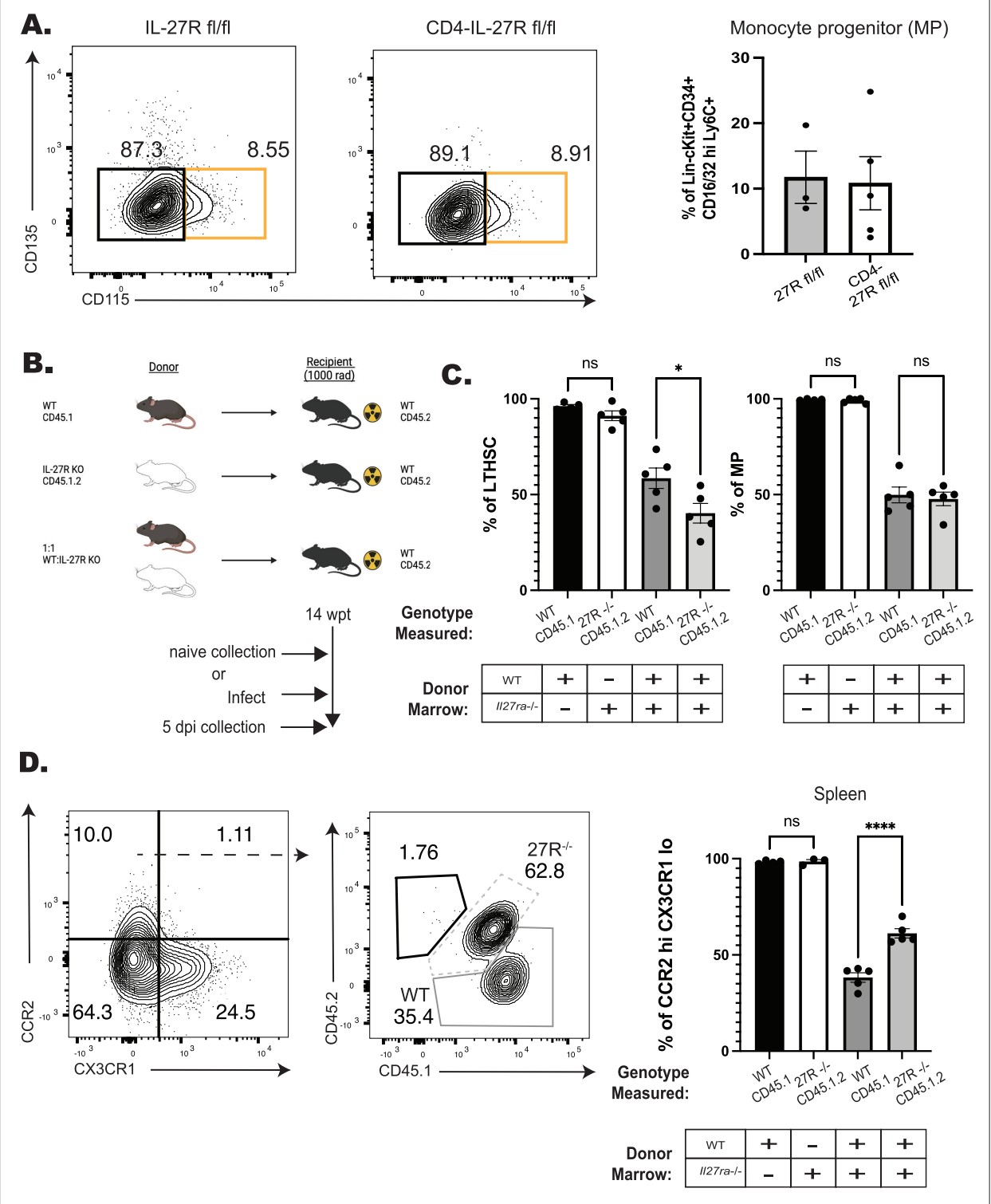

**Figure 2.** IL-27 regulates monopoiesis during infection in a cell-intrinsic manner. (**A**) CD4-IL-27R mice were infected and MPs (orange box) measured at 5 dpi in the BM. Representative flow plots are shown (left) and quantified (right). (**B**) Schematic of BM chimeras generated from WT (CD45.1) and *Il27ra⁻/⁻* (CD45.1.2) bulk bone marrow that were then infected and used in C and D. Schematic created with BioRender.com. (**C**) The proportion of long-term hematopoietic stem cells (LTHSCs) (left) and MPs (right) from each respective donor lineage in the bone marrow of both the single (left two bars) and mixed (right two bars) chimeras at 5 dpi. (**D**) Representative flow plots pre-gated on splenic monocytes (left plot) and then down-gated on CCR2ʰⁱCX3CR1ˡᵒ (right plot) monocytes at 5 dpi. WT-derived monocytes are shown in the solid-dark gray, while *Il27ra⁻/⁻*-derived cells are in the dashed-light gray. The proportion of each donor lineage that contributes to mature CCR2ʰⁱCX3CR1ˡᵒ splenic monocytes at 5 dpi was then quantified (far right).

*Figure 2 continued on next page*

*Figure 2 continued*

Statistical significance was tested by either Welch's *t*-test (**A**) or one-way ANOVA with Sidak's correction (**C, D**). * and **** correspond to p-values ≤0.05 and 0.0001, respectively. *N* = 3–5 mice/group and data shown are representative of two to three repeated experiments. Error bars indcate SEM.

The online version of this article includes the following figure supplement(s) for figure 2:

**Figure supplement 1.** IL-27 regulates monopoiesis during infection and post-irradiation in a cell-intrinsic manner.

the donor (*Figure 2—figure supplement 1B, C*, left bars). In the mixed chimeras, while initially no difference in chimerism was observed, as measured by output of peripheral T and B cells, gradually there appeared to be decreased output from the *Il27ra*$^{-/-}$ lineage (*Figure 2—figure supplement 1B*). This was confirmed in the bone marrow where a reduced number of *Il27ra*$^{-/-}$ LTHSCs and downstream MPs was observed (*Figure 2—figure supplement 1C*, right bars, a 40:60 ratio KO:WT). Despite this reduction in the *Il27ra*$^{-/-}$ LTHSCs, the population of CCR2$^{hi}$CX3CR1$^{lo}$ monocytes in the periphery was preferentially derived from the *Il27ra*$^{-/-}$ lineage (*Figure 2—figure supplement 1D*, right bars, 60:40 ratio KO:WT). To determine if infection would impact this differentiation, these mixed chimeras were infected with *T. gondii* and analyzed at 5 dpi (*Figure 2C*). After infection, *Il27ra*$^{-/-}$ LTHSCs remained deficient in the mixed chimeras, but MPs from both lineages were equivalent (*Figure 2C*, right bars). Additionally, the CCR2$^{hi}$CX3CR1$^{lo}$ monocytes were still predominately derived from the *Il27ra*$^{-/-}$ lineage (*Figure 2D*, right bars). Together, these data indicate that at homeostasis and during infection, IL-27 can support LTHSC populations but also temper the downstream processes that lead to monocyte production. Additionally, this function is independent of IL-27 signals on CD4$^+$ T cells but rather is due to a cell-intrinsic ability of IL-27 to limit the production of inflammatory monocytes.

## IL-27p28 is expressed in the bone marrow at homeostasis and during infection

Given the impact of IL-27 on myelopoiesis described above, studies were performed to characterize the cellular sources of IL-27p28 and their spatial location in the bone marrow of naive and infected mice. WT mice were infected, bone marrow harvested at 0, 3, 5, and 10 dpi, and local production of the IL-27p28 subunit assayed (*Figure 3A*). At 0 dpi, low constitutive levels of IL-27p28 were detected and increased throughout infection, with a peak at 5 dpi and elevated levels maintained at 10 dpi (*Figure 3A*). The use of an *Il27*$^{GFP}$ reporter mouse (*Aldridge et al., 2024*; *Kilgore et al., 2018*) revealed a small population of GFP$^+$ cells in the bone marrow of naive mice, but by 5 dpi, this was expanded (*Figure 3B, C*). Additional analysis of the bone marrow and peripheral tissues (spleen, blood, and peritoneal exudate cell [PEC]) (*Figure 3B, C*) revealed that although a small proportion of T cells were GFP$^+$, consistent with other reports (*Kimura et al., 2016*; *Lin et al., 2023*), monocytes and macrophages were the dominant source of IL-27 in these sites, by proportion of GFP$^+$ cells as well as by cell number (*Figure 3B, C*, *Figure 3—figure supplement 1A*). Additionally, monocyte populations expressed the highest levels of GFP in comparison to other populations, such as dendritic cells (*Figure 3—figure supplement 1B*).

To visualize the spatial relationship between IL-27 expressing cells and HSPCs, femurs from naive *Il-27*$^{GFP}$ mice were harvested and imaged. In naive mice, a stain for Sca-1 revealed low numbers of HSPCs while *Il-27*$^{GFP+}$ cells were present at a higher frequency; however, few of the cells in each population colocalized (*Figure 3—figure supplement 1C*, yellow arrows). During infection, the expression of Sca-1 is upregulated by many cell populations (*Baldridge et al., 2011*; *Morales-Mantilla et al., 2022*), and so the *Procr*$^{creERT2-IRES-tdTomato}$ mouse (identifying HSPCs) was crossed with the *Il-27*$^{GFP}$ reporter. As HSCs are typically found within the hypoxic regions of the bone marrow (*Eliasson and Jönsson, 2010*; *Parmar et al., 2007*), i.v. injection of anti-CD31 was used to label arterioles to distinguish regions of the marrow that are highly vascularized or more hypoxic. Immunofluorescence of the femur from 5 dpi highlighted that, similar to flow cytometric analysis of the BM of the IL-27p28 reporters, infection resulted in increased numbers of *Il27*$^{GFP+}$ cells (*Figure 3D*). *Procr*$^{tdtomato+}$ cells had a homogenous distribution in the femur (*Figure 3—figure supplement 1D*, middle) whereas p28$^{GFP+}$ expression was more regional (*Figure 3D*; *Figure 3—figure supplement 1D*, top). This was quantified by sampling throughout the femur section (*Figure 3D*; *Figure 3—figure supplement 1D*, regions 1–8). The area containing almost exclusively *Procr*$^{tdTomato+}$ HSPCs, but few *Il27*$^{GFP+}$ cells, was an area of low vascularization, consistent with the HSC niche (*Figure 3D*; *Figure 3—figure supplement 1D*,

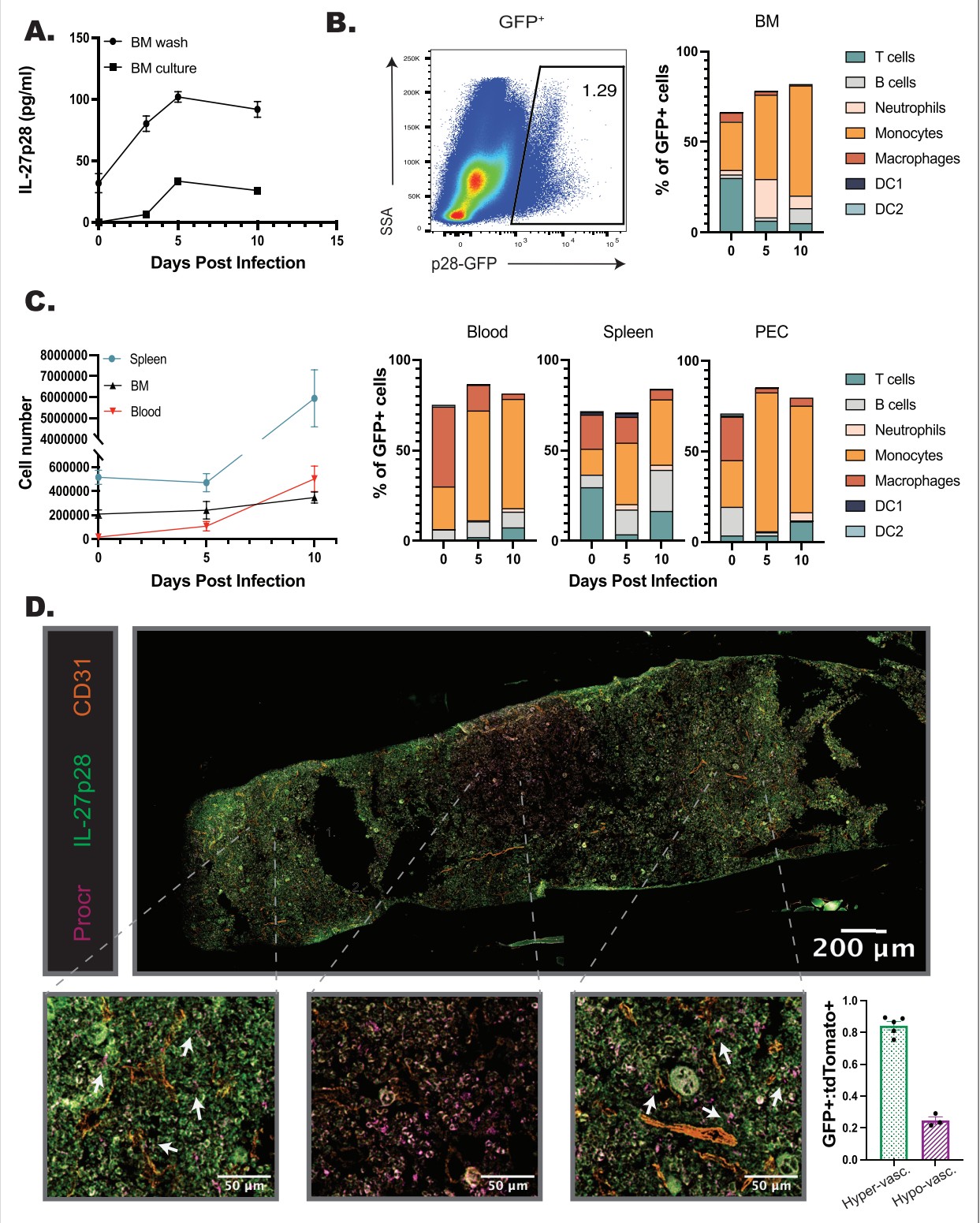

**Figure 3.** IL-27p28 is produced in the BM during infection. (**A**) WT mice were infected, and the bone marrow harvested throughout infection. BM was either washed with media or cultured for 24 hr, and supernatant collected. Both the wash and supernatant were analyzed by ELISA for IL-27p28. (**B**) *Il27*GFP reporter mice were infected and their bone marrow analyzed by flow cytometry throughout infection. A representative flow plot at 5 dpi of bulk marrow is shown (left) and proportion of immune cell contribution to the GFP+ population measured (right). N=3 biological replicates, with two technical replicates performed. (**C**) Numbers of GFP+ cells in the spleen, blood, and BM of infected mice were quantified throughout infection (left).

*Figure 3 continued on next page*

*Figure 3 continued*

Proportions of GFP⁺ CD45⁺ cells were analyzed by cell type in the blood, spleen, and peritoneal exudate cell (PEC) during infection (right). (**D**) Femurs from infected *Procr*^tdTomato+^*Il27*^GFP+^ mice were harvested, sectioned, and imaged at ×40 magnification. The total imaged femur is shown (top), with zoomed-in sections shown below for increased detail of cellular localization (bottom). Arrows indicate areas of localization with *Procr*^tdTomato+^ and *Il27*^GFP+^ cells. Numbers of *Procr*^tdTomato+^ and *Il27*^GFP+^ cells were then counted in eight randomly selected regions throughout the marrow (see **Figure 3—figure supplement 1**). GFP⁺ cells were normalized to tdTomato⁺ cells and the eight counted regions classified as areas of hyper- or hypo-vascularization according to proximity of the region to the labeled vasculature. This allowed quantification of the localization of cells (bottom right). *N* = 3–5 mice/group, and data shown are representative of two repeated experiments. Error bars indcate SEM.

The online version of this article includes the following figure supplement(s) for figure 3:

**Figure supplement 1.** IL-27p28 is produced in the bone marrow during infection.

regions 3–5). However, in regions close to the CD31⁺ vasculature, HSPCs and IL-27p28⁺ cells were found in proximity to one another (**Figure 3D**; **Figure 3—figure supplement 1D**, regions 1–2 and 6–8). When these regions were grouped based on this vascularization, and GFP-expressing cells normalized to tdTomato⁺ cells to account for potential random distribution, a marked decrease in GFP⁺ cells was apparent in hypovascularized regions (**Figure 3D**). These data indicate that during infection, the *Il27*^GFP+^ BM cells associate with HSPCs, and this is most apparent near the vasculature and away from the HSC niche.

## Developmental expression of the IL-27R by hematopoietic cells

The hematopoietic tree begins with undifferentiated LT-HSCs that differentiate into distinct hematopoietic lineages and mature progeny; the color scheme in **Figure 4A** reflects these developmentally related populations. To determine which progenitor stages in the marrow were responsive to IL-27, the expression of the IL-27Rα was compared with other cytokine receptors (gp130, the IL-2R common gamma chain (CD132), IL-6Rα, and GM-CSFRα) that modulate hematopoiesis (**Collins et al., 2021**; **Reynaud et al., 2011**). Expression of these receptors was measured on HSPCs and committed progeny in naive and infected mice (**Figure 4B-C**; **Figure 5A-D**). The gp130 receptor, one chain of the IL-27 receptor that is also utilized by other IL-6 family members, was expressed by all developmental stages examined in both naive and infected mice (**Figure 4B**). In contrast, the IL-27Rα was highly expressed by LTHSCs, but markedly decreased with lineage commitment, with mature neutrophils and monocytes showing no expression (**Figure 4C**). The use of the Haemosphere RNA-seq dataset (**Choi et al., 2019**) and the BloodSpot online tool (**Bagger et al., 2016**) for the analysis of normal mouse hematopoietic bone marrow transcripts (**Chambers et al., 2007**; **Di Tullio et al., 2011**) confirmed that levels of *Il27ra* mRNA were high in LSKs and LTHSCs but decreased with lineage commitment (**Figure 4—figure supplement 1A, B**). To determine if IL-27R signaling was functional on HSPCs, bulk bone marrow was collected from M1Red reporter mice (**Pardy et al., 2024**; **Stifter et al., 2019**), which express RFP under the control of the *Irgm1* promoter, downstream of STAT1 signaling. When these cells were treated with IL-27 (an activator of STAT1), the myeloid-committed progenitors showed minimal RFP expression (**Figure 4—figure supplement 1C**), whereas HSPCs showed a time-dependent increase in reporter expression (**Figure 4D**). This differed dramatically from cells isolated ex vivo from IRGM1 reporter mice treated with another activator of STAT1, the cytokine IFN-γ. After 6 hr, all hematopoietic progenitors analyzed exhibited IRGM1-RFP expression (**Figure 4—figure supplement 1C** and data not shown). The bulk BM cultures stimulated with IL-27 also confirmed that known IL-27R expressing T, B, and NK cells within the culture also responded to IL-27 with a similar kinetic of RFP activation to that of the HSPCs (**Figure 4—figure supplement 1C**).

When expression of the common IL-2R gamma chain (utilized by IL-2, –4, –7, –9, –15, and –21) was analyzed, like gp130, it was ubiquitously expressed by all developmental stages, although infection did upregulate its expression (**Figure 5A**). Similarly, CD131, which is shared by GM-CSF, IL-3, and IL-5 signaling, was expressed in most developmental stages and was markedly enhanced upon infection (**Figure 5B**). In contrast to the IL-27Rα, the expression of IL-6Rα and GM-CSFRα by HSPCs was negligible, and the highest expression of these was detected on fully committed, mature neutrophils and monocytes (**Figure 5C, D**). Additionally, both the IL-6Rα and GM-CSFRα showed enhanced expression during infection, further differing from IL-27Rα. A summary of the survey for IL-10R, IFN-γRβ, IFN-αR, IL-17Rα, TNFIR, and TNFIIR is presented in **Figure 5—figure supplement 1** and highlights that while some of these were constitutively expressed across the hematopoietic tree, none of these

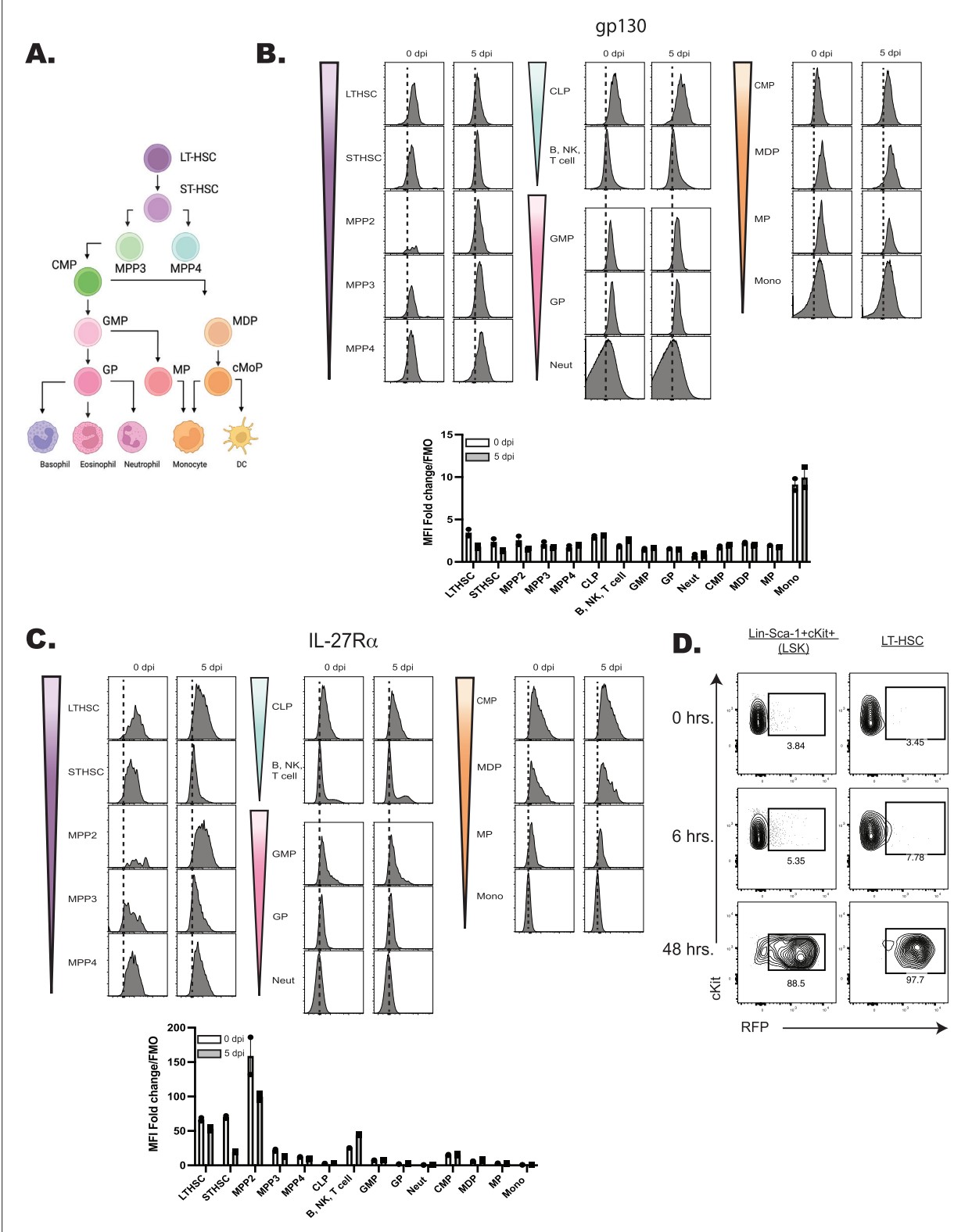

**Figure 4.** Expression of the IL-27R subunits gp130 and IL-27Rα during hematopoietic development. (**A**) Schematic ball-and-stick model of hierarchical hematopoietic development. Cell type colors are maintained for reference in the following panels. (**B**) Progenitors and progeny shown in (**A**) were analyzed in WT and infected mice (at 5 dpi) by flow cytometry for the gp130 receptor. 10,000 live cells were concatenated from *n* = 3 mice, and representative flow plots are shown. The fold change of the MFI of the receptor in each cell type over the FMO control for that cell type was then quantified. Error bars indicate SEM. (**C**) Cells in (**B**) were analyzed for expression of the IL-27Rα and analyzed as above. (**D**) Bulk bone marrow was

*Figure 4 continued on next page*

*Figure 4 continued*

isolated from *Irgm1*[dsRed] reporter mice. 22 × 10[6] cells were plated per condition and stimulated with 20 ng/ml of IL-27 for 0, 6, or 48 hr and RFP expression measured by flow cytometry. Data are representative of two repeated experiments.

The online version of this article includes the following figure supplement(s) for figure 4:

**Figure supplement 1.** IL-27Rα transcript expression during hematopoiesis and differential sensitivity of long-term hematopoietic stem cell (LTHSC) and common myeloid progenitor (CMP) to IL-27.

receptors showed preferential expression by HSPCs. Thus, this survey of cytokine receptor expression highlights that HSPCs but not downstream progenitors are uniquely positioned to respond to IL-27.

## Infection-derived IL-27 restrains a monocytic differentiation program in HSPCs

To determine the impact of infection-driven IL-27 on HSPC differentiation, LTHSCs were sorted from naive or infected WT and *Il27ra*[−/−] mice at 5 dpi (see *Figure 6—figure supplement 1* for gating strategy). These LTHSCs were then cultured in MethoCult and their phenotype and ability to form colonies was quantified by morphology after 5 or 10 days of culture (*Figure 6—figure supplement 2*). As expected, cells from naive mice generated all colony types, but in cultures derived from infected mice, there were reduced granulocyte/monocyte (GM) and granulocyte (G) colonies (*Figure 6—figure supplement 2A*). However, by 10 days post-culture (dpc), there was a slight enhancement in macrophage/monocyte colonies (M) from infected mice. The significant loss of colonies from infected mice, though, made quantification of these colonies difficult. Thus, to better quantify these differences, these cultures were analyzed by flow cytometry to measure expression of canonical developmental markers (see Materials and methods). Dimensional reduction was then performed via UMAP, and cluster formation was analyzed with X-shift to determine the dominant markers associated with individual clusters. When cells were concatenated from all collection timepoints, 14 clusters were delineated (*Figure 6—figure supplement 2B*). These clusters were driven, predominantly, by either collection time (*Figure 6—figure supplement 2B*, top) or proliferative status (*Figure 6—figure supplement 2B*, bottom). Clusters 1–9 contained only samples from 5 dpc, and clusters 1–6 corresponded to undifferentiated, quiescent colonies that were minimally proliferative (Ki67[lo]), while clusters 7–9 were starting to proliferate (Ki67[hi]) and differentiate to give rise to mature cell populations. By 10 dpc, there was a loss of the quiescent colonies and a rise of populations (clusters 10–14) that are Ki67[hi] but have increased expression of mature myeloid markers (CD11b, CD16/32, and Ly6C) (*Figure 6—figure supplement 2C*). This reflects the increased differentiation of colonies into mature myeloid cells. The UMAP analysis highlighted clusters 1, 6, 7, and 14 as the dominant output of this assay (*Figure 6—figure supplement 2D*): Cluster 1-undifferentiated progenitors (Ki67[lo], BCL2[mid], no other expression), Cluster 6—myeloid biased HSPCs (CD150[hi]), cluster 7—monocyte progenitors (intermediate expression of myeloid markers (CD16/32[lo]), but expression of progenitor markers (CD34[int])), and cluster 14—mature monocyte (CD11b[hi], Ly6C[hi], and CD16/32[int]) (*Figure 6—figure supplement 2C*). Furthermore, this clustering analysis highlighted that infection resulted in a loss of myeloid progenitors (cluster 7) that was further reduced in the absence of IL-27 (*Figure 6—figure supplement 2E*). However, by 10 dpc, this corresponded to enhanced differentiation into mature monocytes (cluster 14).

To compare the ability of HSPCs from WT or *Il27ra*[−/−] mice to generate monocyte populations, cells solely from 10 dpc were clustered (*Figure 6A*), with the dominant clusters being 1, 2, 3, 10, and 11 (*Figure 6B*). Of these, HSPCs were present in cluster 1 (Sca-1[+]cKit/CD117[+]), while cluster 10 contained MPs (CD11b[hi]Ly6C[low]CD34[+]CD115[+]), and cluster 11 expressed markers of mature monocytes (CD11b[hi]Ly6C[+]) (*Figure 6C, D*). Further analysis (*Figure 6E*) revealed that HSPCs from infected mice produced increased numbers of monocytes and macrophages (cluster 11), but only HSPCs from infected, *Il27ra*[−/−] mice were enhanced in their differentiation into MPs (cluster 10). Thus, during infection, the loss of IL-27 signaling enhances the skewing of HSPCs ex vivo toward monocyte differentiation.

In a complementary in vivo approach, single transfers of WT or *Il27ra*[−/−] HSPCs from naive or infected mice were transferred into irradiated WT hosts and differentiation assessed (*Figure 6F*). For these experiments, bulk marrow was utilized because, as discussed above, inflammation results in upregulation of Sca-1 and other lineage-defining markers (*Baldridge et al., 2011*; *Morales-Mantilla et al., 2022*), making it difficult to define true HSPCs for transfer by cell sorting. In addition, for these

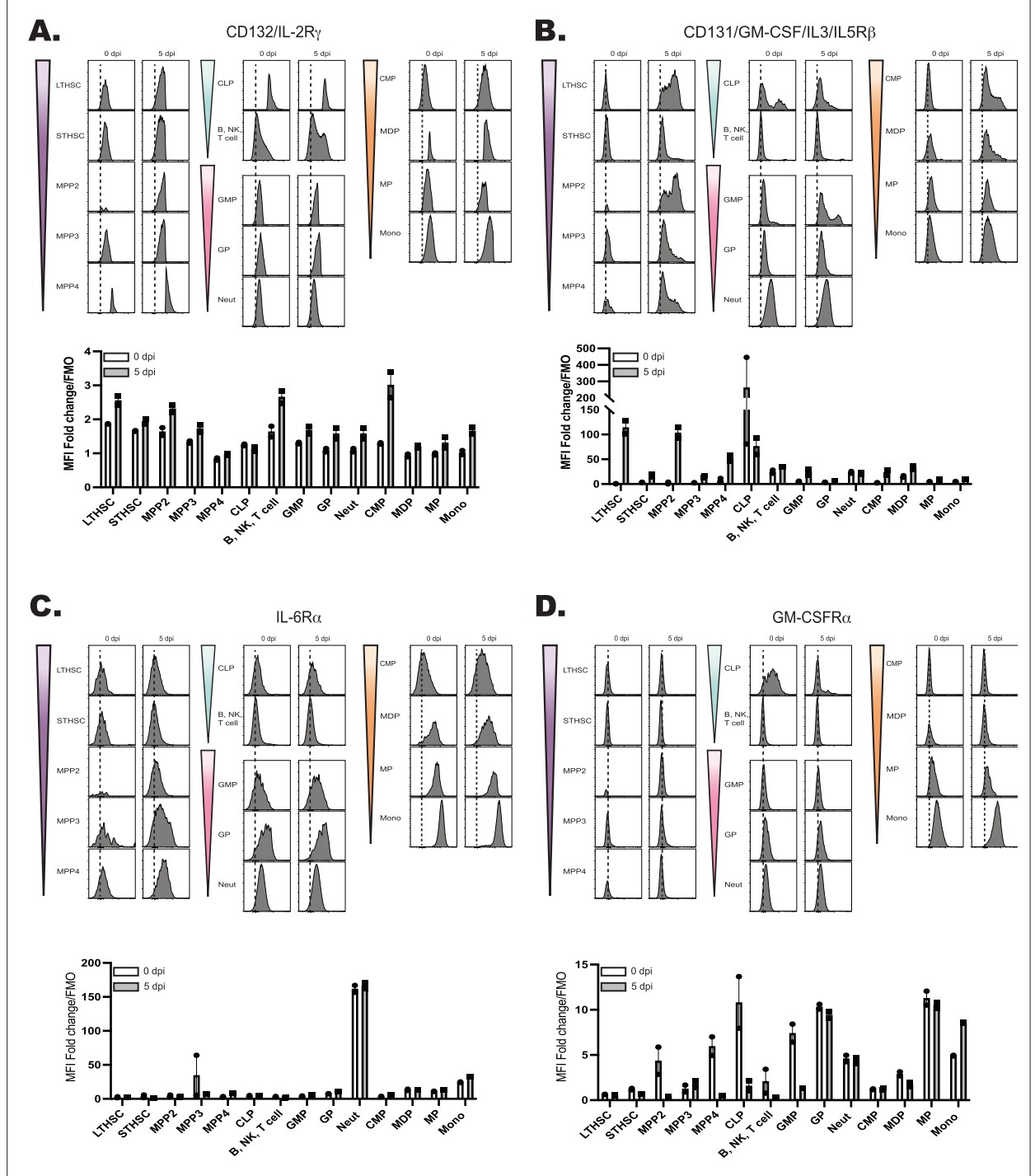

**Figure 5.** Developmental expression of hematopoietic cytokine receptors. (**A**) Expression of CD132 (IL-2Rγ) and (**B**) CD131 (GM-CSF/IL-3/IL-5Rβ) was analyzed by flow cytometry on progenitors and progeny in the BM of naive and 5 dpi mice as in *Figure 4*. (**C**) Expression of IL-6Rα and (**D**) GM-CSFRα was measured as in (**A**).

The online version of this article includes the following figure supplement(s) for figure 5:

**Figure supplement 1.** Developmental expression of additional cytokine receptors.

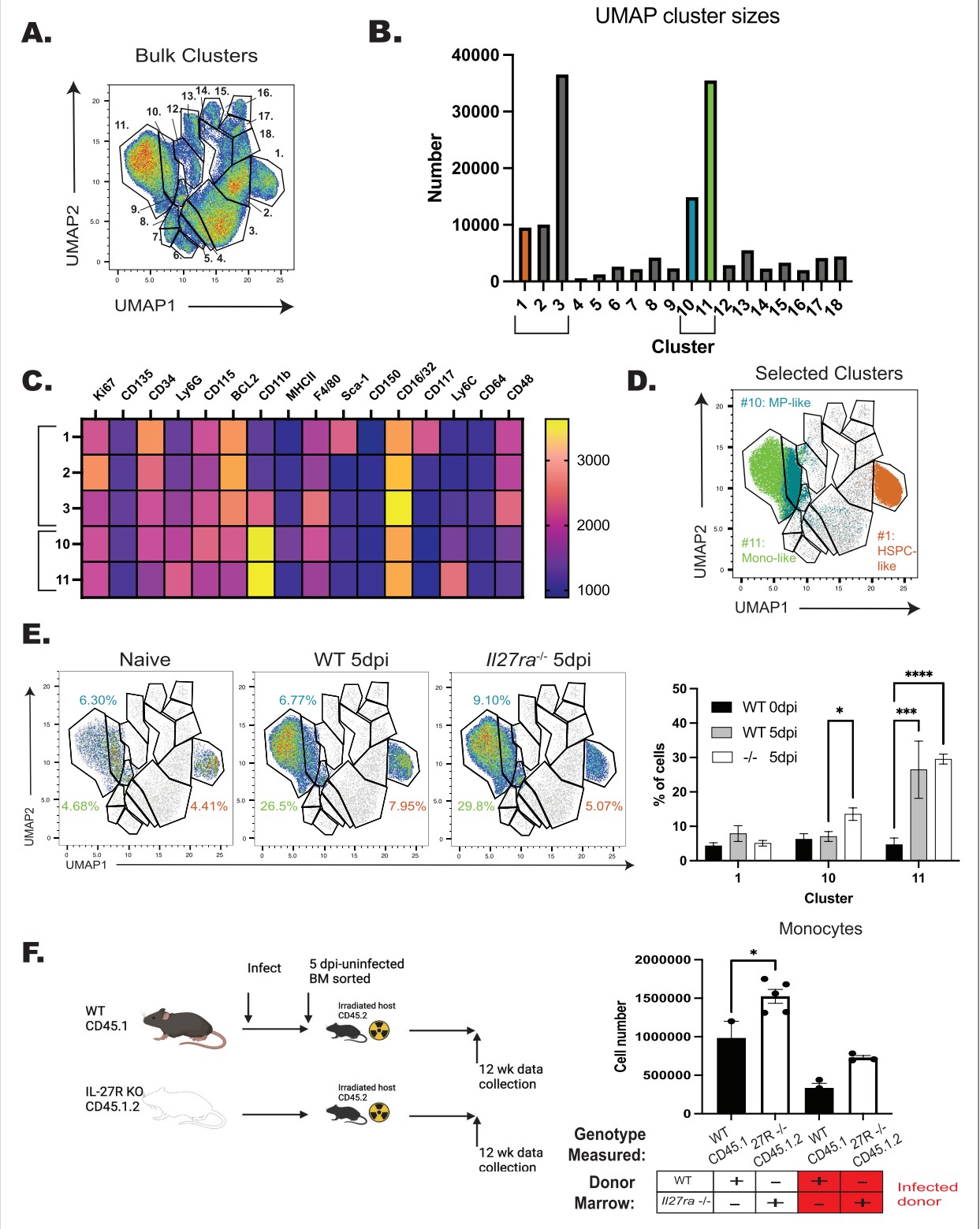

Figure 6. IL-27 limits infection-induced hematopoietic stem and progenitor cell (HSPC) polarization. WT and *Il27ra⁻/⁻* mice were infected for 5 days, long-term hematopoietic stem cells (LTHSCs) sorted, and cultured in MethoCult for 10–12 days before being analyzed by flow cytometry. (**A**) 10,000 live cells from *n* = 3–4 mice/group (WT (naive +5 dpi) and *Il27ra⁻/⁻* (5 dpi)) were concatenated and dimensionally reduced via UMAP. (**B**) Numbers of cells in each cluster from (**A**) were measured. (**C**) X-shift analysis was used to identify the expression level of each marker used in the clustering performed in (**A**). Five of the dominant clusters from (**B**) are shown. (**D**) Clusters 1, 10, and 11, all involved in monocyte development, are highlighted. (**E**) Contribution

*Figure 6 continued on next page*

*Figure 6 continued*

of clusters 1, 10, and 11 to each genotype is shown (left) and quantified (right). (**F**) Single bone marrow chimeras were generated with bone marrow from infected WT and *Il27ra*$^{-/-}$ mice or naive controls (left). The numbers of single-donor-derived, mature monocytes were then measured at 14 weeks post-transplant and compared based on WT vs *Il27ra*$^{-/-}$ donor (right). Statistical significance was tested by one-way ANOVA with Sidak's correction. \*, \*\*\*, and \*\*\*\* correspond to p-values ≤0.05, 0.001, and 0.0001, respectively. *N* = 3–5 mice/group, and data shown are representative of two repeated experiments. Error bars indicate SEM. Schematic in (**F**) created with BioRender.com.

The online version of this article includes the following figure supplement(s) for figure 6:

**Figure supplement 1.** Gating strategy for sorting of long-term hematopoietic stem cells (LTHSCs) used in monocyte differentiation.

**Figure supplement 2.** Development and validation of an unbiased flow cytometry approach for the analysis of MethoCult colonies.

studies, mice were infected with a strain of *T. gondii* that expresses tdTomato (Pru-tdTomato) (*Christian et al., 2014*; *John et al., 2009*) to allow parasite removal during sorting and prevent infection of irradiated recipients. Analysis of spleens of recipient mice at 12 weeks revealed that donors from naive or infected *Il27ra*$^{-/-}$ mice produced a greater number of monocytes than those derived from WT mice (*Figure 6F*). These ex vivo and in vivo approaches indicate that exposure of LTHSCs to IL-27 during infection results in a stable program that limits monocyte production.

## IL-27 regulates the functionality and fitness of HSPCs during infection

While the datasets presented above indicate a role for IL-27 in the regulation of HSPC differentiation, it was unclear if loss of IL-27 signals during infection would affect their functionality and fitness. To test if there were alterations in HSPC proliferation during infection, WT and *Il27ra*$^{-/-}$ mice were treated with BRDU throughout infection, and HSPC proliferation was analyzed at 5 dpi. The combination of Ki67 expression and BRDU incorporation was used to distinguish HSPCs entering cell cycle (Ki67$^+$BRDU$^-$), cells that have entered cell cycle and are actively proliferating (Ki67$^+$BRDU$^+$), and those that have proliferated but are no longer in cell cycle (Ki67$^-$BRDU$^+$). In WT mice, the majority of LSKs are quiescent (Ki67$^-$BRDU$^-$), a proportion of cells have entered cell cycle (Ki67$^+$ BRDU$^-$), and a distinct Ki67$^+$BRDU$^+$ population, as well as a smaller Ki67$^-$BRDU$^+$ population, can be detected. In the absence of IL-27 signaling, the most prominent difference was a reduced proportion of Ki67$^+$BRDU$^+$ cells (*Figure 7A*). To determine if this decreased Ki67$^+$BRDU$^+$ population might be due to increased cell death of HSPCs in the IL-27R$^{-/-}$ mice, cells were stained for Annexin V and a free-amine reactive viability dye to discriminate early- and late-apoptotic cells as well as necrotic cells. No difference in cell death was observed between WT and *Il27ra*$^{-/-}$ HSPCs (*Figure 7—figure supplement 1B*). These data need to be interpreted with care, as total HSPC numbers are not different between WT and *Il27ra*$^{-/-}$ mice (see *Figure 1*), but they suggest that IL-27 may regulate either the proliferation or retention of HSPCs within the bone marrow niche.

A key feature of healthy HSCs is their ability to reconstitute the hematopoietic system of irradiated mice, even after serial transfer (*Siminovitch et al., 1964*; *Weissman, 2000*). However, inflammation and over-proliferation of HSCs can impair their ability to rescue lethally irradiated recipients (*Feng et al., 2008*; *King et al., 2011*; *Pietras et al., 2014*). To test the impact of IL-27 on HSC fitness and functionality, serial transplantations of HSCs from infected WT and *Il27ra*$^{-/-}$ mice were performed. In these experiments, marrow from naive or infected donors was used to produce single chimeras or to provide a 1:1 mix of WT and *Il27ra*$^{-/-}$ populations that were transferred to irradiated recipients (*Figure 7B*). All transfers promoted host survival and at 9 weeks post-transplantation, the WT or *Il27ra*$^{-/-}$ donors from naive and infected mice reconstituted the blood compartment with a similar efficiency (*Figure 7—figure supplement 1C*). However, in the 1:1 chimeras from infected mice, the peripheral compartment was dominated by cells derived from WT mice. Analysis of the marrow of the single transfer chimeras at 12 weeks post-transplant highlighted that when donor cells were derived from infected mice, there was a significant reduction in the numbers of LTHSCs (*Figure 7C*). To test the functionality of these HSCs, they were then transplanted for a second time into irradiated hosts (*Figure 7D*). Mice that received a secondary transplant of marrow from naive WT and *Il27ra*$^{-/-}$ mice survived irradiation, consistent with the ability of healthy HSCs to undergo at least three rounds of transplantation (*Siminovitch et al., 1964*; *Weissman, 2000*). Additionally, secondary transplantation of WT HSCs from infected mice also resulted in survival of these hosts. However, the serial transfer of marrow from infected *Il27ra*$^{-/-}$ mice did not save irradiated recipients (*Figure 7D*). Thus, while

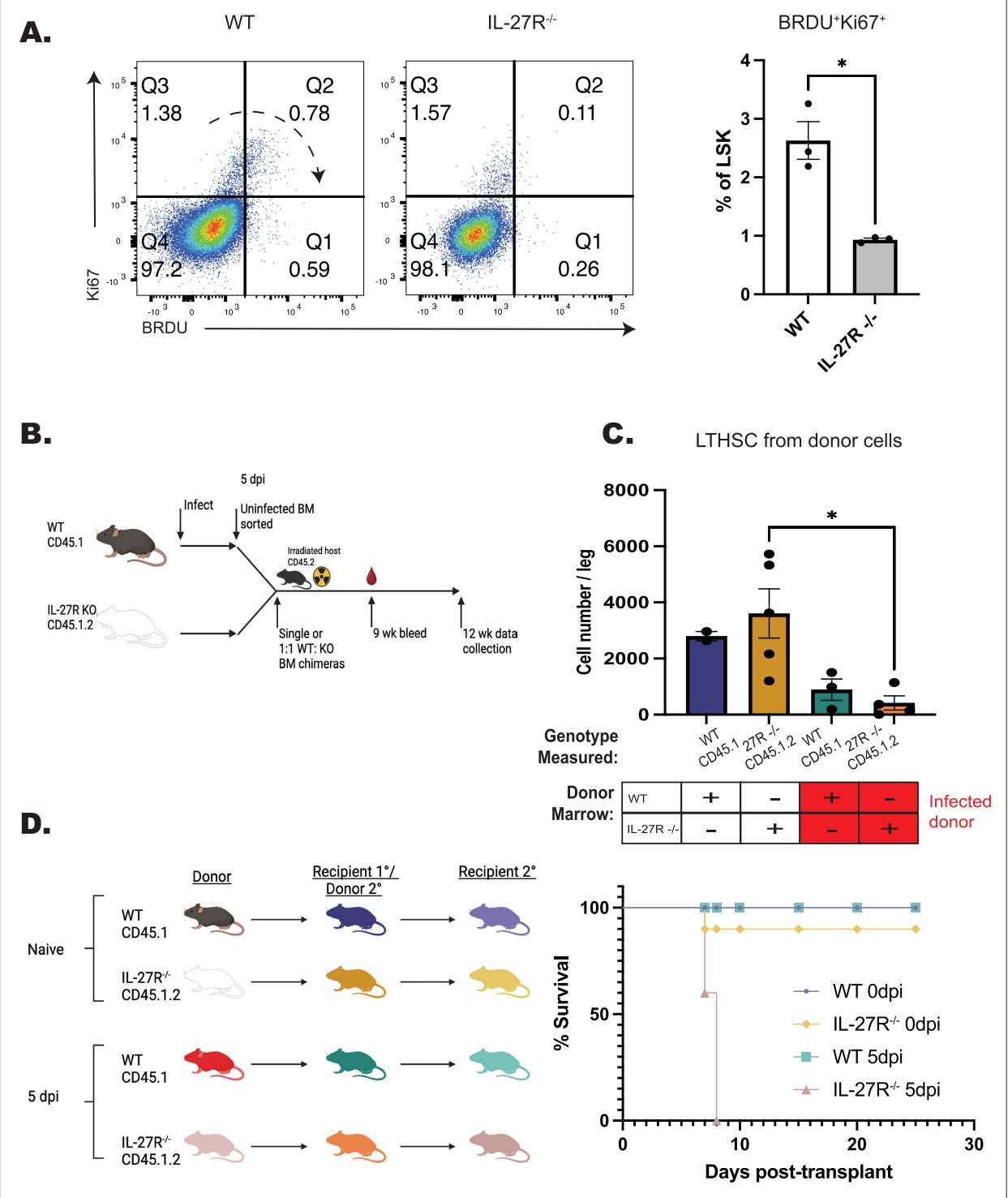

**Figure 7.** Testing the impact of IL-27 on functionality and fitness of hematopoietic stem and progenitor cells (HSPCs) during infection. (**A**) WT and *Il27ra*⁻/⁻ were infected, treated with 5 mg BRDU, and kept on 1 mg/ml BRDU in the drinking water throughout infection. Incorporation of BRDU was then measured in Lin⁻Sca-1⁺cKit⁺ (LSK) progenitor cells in comparison to Ki67 at 5 dpi. Representative flow plots are shown (left) and the proportion of BRDU⁺Ki67⁺ cells quantified (right). (**B**) Schematic for the generation of BM chimeras from infected mice used in (**C**), schematic created with BioRender. com. (**C**) As in **Figure 6F**, numbers of long-term hematopoietic stem cells (LTHSCs) from recipients receiving single-transfer donor marrow from either uninfected or infected mice were measured based on genotype. (**D**) Secondary transplant chimeras from mice that originally received naive or infected marrow were generated as shown (left) image created with BioRender.com. Survival of recipients was then assessed (right). Statistical significance was

*Figure 7 continued on next page*

*Figure 7 continued*

tested by Welche's t-test (A) or one-way ANOVA with Sidak's correction (C). * corresponds to p-values ≤0.05. *N* = 3–5 mice/group, and data shown are representative of two experiments. Error bars indicate SEM.

The online version of this article includes the following figure supplement(s) for figure 7:

**Figure supplement 1.** Testing hematopoietic stem and progenitor cell (HSPC) survival and fitness.

*Il27ra⁻/⁻* HSPCs from infected mice maintain a differentiation bias toward the monocyte lineage, they have a decreased survival and proliferation capacity and a reduced ability to rescue irradiated mice.

## Discussion

During toxoplasmosis, the increased production of inflammatory monocytes is critical to control infection (*Dunay et al., 2008*), but these populations also have regulatory properties that help to limit immune pathology. For example, as infection progresses, the presence of IFN-γ in the bone marrow promotes the production of monocytes that produce IL-10 and $PGE_2$ (*Askenase et al., 2015*), which contribute to the restoration of homeostasis. Similarly, during toxoplasmosis and other infections in response to IFN-γ/STAT1 signals (*Furusawa et al., 2016* and our own unpublished observations) monocytes become a dominant source of IL-27 (*Detavernier et al., 2019*; *Hall et al., 2012*). While IL-27 has a critical role in directly limiting pathological T cell responses, the data presented here show that IL-27 is also part of a regulatory loop that acts on HSPCs to limit emergency myelopoiesis. Infection-induced monopoiesis gives rise to a population of IL-27⁺ monocytes in the bone marrow that can then act on HSPCs to limit further monocyte production. This observation is reminiscent of reports that IFN-γ-activated monocytes/macrophages can directly limit HSPCs (*McCabe et al., 2015*; *Seyfried et al., 2020*) or the ability of IL-10 to limit emergency myelopoiesis in the fetus (*Collins et al., 2024*). Likewise, during infection with *T. gondii*, Tregs (*Glatman Zaretsky et al., 2017*) appear to participate in a regulatory cross-talk with HSPCs. A better understanding of how immune cells in the bone marrow, such as memory B and T, influence HSPCs is needed.

One challenge that faces HSCs during inflammation is the need to protect from stressors associated with proliferation that adversely affect the functions of these cells (*Caiado et al., 2021*). The concept of HSC 'exhaustion' has been developed to describe HSCs that are no longer able to fulfill their role in proliferation and reconstitution of steady-state hematopoiesis (*Singh et al., 2020*; *Zhao and Deininger, 2023*). Unanticipated findings from our studies here were the reduced ability of *IL27ra⁻/⁻* HSCs post-infection to compete with WT HSCs, as well as the inability of these HSCs to reconstitute upon secondary transfer, hallmarks of functional exhaustion (*Singh et al., 2020*). These results infer that the ability of IL-27 to limit emergency myelopoiesis is also associated with the ability of IL-27 to limit HSC exhaustion. It is important to note, however, that these studies indicate that multiple cells within the HSPC category, including MPPs, express the IL-27Rα and are responsive to IL-27. Whether the impacts on HSC exhaustion detected here are due to the direct effects of IL-27 on HSCs or on downstream MPPs, then, warrants further investigation. Aberrant HSPC responses are associated with conditions such as clonal hematopoiesis and aging, and there has been interest in the development of strategies to target these HSPCs. Thus, selective targeting and depletion of 'aged' HSPCs via anti-CD150 treatments has been reported to restore balanced hematopoietic output and 'youthful' blood phenotypes (*Ross et al., 2024*). Whether IL-27 treatment can be used to preserve HSPCs, or be blocked to enhance monocyte production, remains to be tested.

A common feature of certain models of trained immunity that affect hematopoiesis is that cytokines, such as IFN-γ and IL-6, promote long-term changes that polarize HSPCs toward myeloid differentiation (*Cheong et al., 2023*; *Kaufmann et al., 2018*; *Khan et al., 2020*). Inherent in this process is the idea that HSCs or downstream progeny would be sensitive to different cytokines. The ability to survey the expression of cytokine receptors that influence hematopoiesis emphasized that the shared cytokine receptor chains gp130 and IL-2Rγ are uniformly expressed throughout hematopoietic development. In contrast, the alpha chain receptors for individual cytokines (IL-6Rα, GM-CSFRα, IL-10R, IFN-γRβ, IFN-αR, IL-21R, TNFRI, and TNFRII) displayed a much more restricted pattern of expression that correlated with key features of the biology of individual cytokines. For example, the limited expression of the IL-6 receptor on downstream progeny is consistent with reports of its high expression on MPPs but not on HSCs (*Mirantes et al., 2014*; *Reynaud et al., 2011*). Nevertheless,

that HSCs express high levels of the IL-27R, which decreases with differentiation, emphasizes that these stem cells are especially sensitive to the effects of IL-27. Whether IL-27 tempers some of the processes associated with innate training or is itself associated with long-term epigenetic changes is the subject of ongoing studies.

The use of murine models of toxoplasmosis has revealed many of the inhibitory activities of IL-27, and the datasets presented here highlight the ability of IL-27 to influence HSPCs to restrain monocyte/macrophage induction. There are also reports during other infections and models that show IL-27 can regulate neutrophil and monocyte development (*Furusawa et al., 2016*; *Liu et al., 2014*; *Liu et al., 2021*; *Seita et al., 2008*; *Sun et al., 2017*; *Wirtz et al., 2006*). For example, during infection with an attenuated strain of *Plasmodium berghei*, IL-27 was required for the induction of a neutrophil response, and in vitro IL-27 blunted the ability of HSPCs to differentiate into the macrophage lineage (*Furusawa et al., 2016*). However, in less inflammatory settings, such as tumor growth, the overexpression of IL-27 resulted in LSK expansion and the induction of an M1 macrophage population (*Zhu et al., 2023*). Likewise, in a model of angiotensin-mediated atherosclerosis, IL-27 was needed to overcome HSC quiescence and increase differentiation and output of mature myeloid cells (*Peshkova et al., 2019*). While it can be difficult to ascribe some of these effects to IL-27 acting directly on HSC, the contrasting nature of these datasets suggests that the impact of IL-27 on HSPCs is context dependent. Indeed, reports that TNFα can upregulate HSPC expression of the IL-27R (*He et al., 2020*) suggest that IL-27 responsiveness can be regulated by the local environment. A better appreciation of these environmental signals at homeostasis and during different types of inflammation, then, should help to define these context-dependent activities of IL-27 on hematopoiesis.

## Materials and methods
### Mice and infection
Female and male C57BL/6 mice were purchased (8–12 weeks old; strain # 000664; RRID:IMSR_JAX:000664) from Jackson labs as age and sex matched controls for whole body knockout mice. CD45.1 donor mice were either bred in-house or purchased from Jackson labs (10–12 weeks old; strain # 033076; RRID:IMSR_JAX:033076). CD45.2 recipient mice were purchased from Taconic labs (10–12 weeks old; model # B6-M; RRID:IMSR_TAC:B6). *Il27ra*⁻/⁻ (Jackson labs; strain # 018078; RRID:IMSR_JAX:018078) were bred in house, as were *Il27*⁻/⁻bio mice, originally generated by Lexicon Pharmaceuticals, Inc and previously described (*Hall et al., 2012*). *Il27*^GFP reporters (*Kilgore et al., 2018*) were similarly bred in house, and the Procr-Ai6 mouse was generated from crossing the *Procr*^creERT2-IRES-tdTomato line (Jackson labs; strain # 033052; RRID:IMSR_JAX:033052), which had been backcrossed nine generations onto the C57BL/6 background and was provided by Dr. Nancy Speck, to the Ai6 mouse line (Jackson labs; strain# 007906; RRID:IMSR_JAX:007906). Mice that contain a floxed allele of the *Il27ra* (*Il27ra*^fl/fl) were provided by Dr. Booki Min (*Do et al., 2017*) and maintained within our mouse colony. This line was then crossed with a *Cd4*^cre mouse (Jackson labs; strain# 022071; RRID:IMSR_JAX:022071) to generate the *Cd4*^cre-*Il27ra*^flox/flox mouse line. M1Red mice, which express red fluorescent protein (RFP, encoded by *dsRed2*) under control of the *Irgm1* promoter and have been previously described (*Stifter et al., 2019*), were received from Dr. Gregory Taylor (Duke University) and maintained in-house. All mice were housed and bred in specific pathogen-free (SPF) facilities in the Department of Pathobiology at the University of Pennsylvania in accordance with institutional guidelines (IACUC# 805045). Cysts of the ME49 strain of *T. gondii* were collected from chronically infected CBA mice brain tissues. Experimental mice were then infected i.p. with 20 cysts. For experiments where uninfected bone marrow was sorted, a Prugniaud (PRU) strain of *T. gondii* expressing tdTomato was used (Pru-tdTomato). Both of these are type II strains and show similar virulence. Inclusion criteria for experimental groups during infection were mice exhibiting signs of infection (i.e. weight loss) and increased immune cell infiltration into tissues as well as bone marrow expansion, as measured by flow cytometry. For experiments including naive controls, mice were deemed naive if uninfected and not exhibiting any sign of abnormal immune cell enhancement, as detected during flow cytometric analysis. Under these criteria, no mice were excluded from our experiments. Each experiment was repeated at least twice and with at least *n* = 3 mice/group. Exact numbers are specified in the figure legends below. The experimental cohorts were not blinded from the researchers.

## Functional analysis of HSPC in vivo

For induction of the *Procr*[creERT2-IRES-tdTomato], tamoxifen was given for three consecutive days (8 mg/ mouse) prior to infection. For BRDU proliferation labeling, 5 mg of BRDU was given i.p. to each mouse the same day as infection. 1 mg/ml of BRDU was then provided in the drinking water to mice throughout infection. To generate bone marrow chimeras, recipient mice were given a radiation dose of 900–1000 rads at least 6 hr prior to transferring donor marrow. Donor marrow was transferred (at least $2 \times 10^6$ cells from each donor, unless otherwise stated; this resulted in a total of $4 \times 10^6$ cells being transferred for mixed chimeras) intravenously (i.v.) via retro-orbital injection. Mice were then given 5 ml of Sulfamethoxazole/Trimethoprim (200 mg + 40 mg/5 ml) in their drinking water and replaced every 2–3 days for 2 weeks.

## Functional analysis of HSPC in vitro

C57BL/6 mice were infected with 20 cysts of ME49 and LTHSCs (Lin-Sca-1+cKit + CD135-CD48-CD150+; see *Figure 6—figure supplement 1*) sorted from the BM of infected and uninfected controls using a FACSAria cell sorter (BD). 400 LTHSCs were then plated in MethoCult GF M3434 (Stem Cell technologies; Cat # 03434) according to the manufacturer's instructions and harvested at either 5 or 10 days of culture for flow cytometric analysis (see below).

## Cell staining, flow cytometry, UMAP analysis, and detection of IL-27

Splenocytes were obtained by grinding spleens through a 70-µm filter and washing with RPMI supplemented with 5% fetal bovine serum (FBS). Red blood cells were then lysed by incubating samples with ACK lysis buffer (Thermo Fisher Scientific) for 5 min before washing again with RPMI + 5% FBS. Both femurs and tibias were harvested from mice for bone marrow isolation. Connective tissue was removed from the bones by gentle scraping with a scalpel before one end of each bone was cut off to expose the marrow. Bones were then placed in a 0.5-ml microcentrifuge tube, with a hole punctured in the bottom with a 31-G needle. The 0.5 ml tube was then placed within a 1.5-ml microcentrifuge tube; the nested tubes were then spun down at 13,000 rcf for 1.5 min. The pelleted cells were then ACK lysed and washed as above (centrifuging at 500 rcf) before being passed through a 70-µm strainer and resuspended in appropriate media (adapted from *Amend et al., 2016*). For PECs, 8 ml of PBS was injected into the peritoneum of mice (21 G needle attached to a 10-ml syringe) and re-aspirated using the same needle and syringe. Syringe contents were then expelled into a 15-ml conical tube, spun down at 300 rcf, and resuspended in appropriate media.

After cell numbers were determined for each sample, equivalent numbers of cells were plated in a 96-well plate and spun down at 300 rcf. Cells were then washed with FACS Buffer [1x PBS, 0.2% bovine serum antigen, 1 mM EDTA] before incubating with Fc block [99.5% FACS Buffer, 0.5% normal rat serum, 1 µg/ml 2.4G2 IgG antibody] prior to staining. If CD16/32 was stained for, the staining antibody was substituted for unconjugated 2.4G2 antibody in the Fc block. Cells were stained with the viability dye Ghost Dye Violet 510 (Tonbo Biosciences; 12-0870), and the following antibodies were used for subsequent staining: CD90.2 (BD; clone 30-H12), B220 (BD; clone RA3-6B2), Ly6G (BD; clone 1A8), CD11b (BD; clone M1/70), Ly6C (Biolegend; clone HK1.4), Sca-1 (BD/Invitrogen; clone D7), CD117/cKit (Biolegend; clone ACK2), CD135 (Biolegend; clone A2F10), CD150 (Biolegend; clone TC15-12F12.2), CD48 (Biolegend/BD; clone HM48-1), CD127 (Biolegend; clone A7R34), CD16/32 (BD; clone 2.4G2), CD34 (Invitrogen; clone RAM34), CD115 (Biolegend; clone AFS98), CCR2 (Biolegend; clone SA203G11), CX3CR1 (Biolegend; clone SA011F11), CD11c (Biolegend; clone N418), XCR1 (Biolegend; clone ZET), CD172a (Biolegend; clone P84), CD64 (Biolegend; clone X54-5/7.1), F4/80 (Invitrogen; clone BM8), MHCII (Biolegend; clone M5/114.15.2), gp130 (Biolegend; clone 4H1B35), IL-27Rα (BD; 2918), CD132 (BD; clone 4G3), CD131 (BD; clone JORO50), IL-6Rα (Biolegend; clone D7715A7), GM-CSFRα (R&D Systems; clone 698423), CD210/IL-10R (Biolegend; clone 1B1.3a), IFN-γR β (Biolegend; clone MOB-47), IFNAR-1 (Biolegend; clone MAR1-5A3), IL-21R (Biolegend; clone 4A9), CD120a/TNFRI (Biolegend; clone 55R-286), CD120b/TNFRII (Biolegend; clone TR75-89), and CD217/ IL-17RA (Biolegend; clone S19010F).

Intracellular staining for Ki67 (BD; clone B56), and BCL2 (Biolegend; clone BCL/10C4) was performed using the Foxp3/Transcription Factor Staining Buffer Set (eBioscience) according to the manufacturer's instructions. BRDU staining was done following a previously published protocol (*Matatall et al., 2018*). Cells within the vasculature were labeled by injecting 3 µg of fluorescent

anti-CD45 antibody (Biolegend; clone 30-F11)/mouse for 3 min prior to euthanasia. Samples were run on a FACSymphony A3 or A5 (BD) and analyzed using the FlowJo Software analysis program (TreeStar; RRID:SCR_008520). For UMAP analyses, either 500 or 10,000 live cells (for 5 days of culture or 10, respectively) per sample were sampled using the downsampling plug-in (TreeStar) and subsequently concatenated. The concatenated samples were then used for UMAP analysis with the FlowJo plug-in (TreeStar), using Euclidean distances with the nearest neighbor set to 15, minimum distance of 0.5, and number of components set to 2. All compensated parameters were then used in the analysis. After the UMAP analysis was performed, the X-shift plug-in (TreeStar) was used to further analyze each cluster for its defining features (*Samusik et al., 2016*).

For IL-27p28 detection, mouse IL-27p28/IL-30 Quantikine ELISA kit (M2728; R&D Systems) was used according to the manufacturer's protocol.

## Immunofluorescence of bone marrow and image analysis

*Procr*$^{tdTomato}$*Il27*$^{GFP}$ mice were generated by crossing the *Procr*$^{creERT2-IRES-tdTomato}$ and *Il27*$^{GFP}$ mouse lines. These mice were then infected, and long bones were harvested at 5 dpi. The bones were then fixed in 4% PFA for 2.5–4 hr, washed with PBS 3x, and then decalcified in 0.5 mM EDTA for ≥72 hr before being paraffin embedded. 5 µm sections were taken and placed onto charged microscope slides. For staining, slides were de-paraffinized, rehydrated, and placed in 1 mM EDTA (pH 8.0) overnight in a 55°C water bath to recover epitopes prior to staining. Slides were washed in diH$_2$O the next day before proceeding with staining. Staining was adapted from *Im et al., 2019*. Slides were blocked with 100 µl of blocking buffer (1x PBS + 0.3% Triton X-100+1:200 rat IgG) for 30 min at room temperature. Slides were then washed 3x, for 5 min each with PBS. Endogenous streptavidin and biotin were then blocked using the ReadyProbes Streptavidin/Biotin Blocking solution (Invitrogen) according to the manufacturer's directions. After washing, GFP was stained for at 1:40 with anti-GFP AF488 (Biolegend; clone FM264G) overnight at 4°C. Slides were washed 3x with PBS, as above, and then mounted and coverslipped using 15 µl Fluoro-Gel (Electron Microscopy Sciences). Slides were then imaged on a Stellaris Falcon (Leica) microscope.

Images were processed and analyzed using the Fiji software package (*Schindelin et al., 2012*) (RRID:SCR_002285). Images were converted to binary images, manually thresholded, watersheding applied to separate overlapping objects, and positive cells counted using the 'analyze particles' tool.

## Statistics

An unpaired, Welch's *t*-test or one-way ANOVA with Sidak's correction for multiple comparisons was used to test for significant differences between groups (GraphPad Prism; RRID:SCR_002798). These were used as indicated in each figure and unless otherwise stated. p-values of less than 0.05 were considered significant.

## Acknowledgements

We would like to thank Dr. Nancy Speck for generously providing us with the *Procr*$^{cre}$ mouse line. Additionally, we thank Dr. Gordon Ruthel for his assistance and instruction in capturing images of the mouse bone marrow. Schematics were created using BioRender.com (agreement license BU27RC-NPBH). Funding for this work was provided by the NIH equipment grant (S10 OD032305-01A1), NIH fellowship support (F31 AI 161962-03), and the Emmerson Collective.

## Additional information

### Funding

| Funder | Grant reference number | Author |
| --- | --- | --- |
| National Institute of Allergy and Infectious Diseases | F31 AI 161962-03 | Daniel L Aldridge |
| National Institutes of Health | U01 AI160664 | Christopher A Hunter |

| Funder | Grant reference number | Author |
| --- | --- | --- |
| National Institutes of Health | R01 AI157247 | Christopher A Hunter |
| National Institute of Allergy and Infectious Diseases | F31 AI179240 | Zachary Lanzar |
| National Institute of Allergy and Infectious Diseases | R01 AI125247 | Booki Min |

The funders had no role in study design, data collection, and interpretation, or the decision to submit the work for publication.

## Author contributions

Daniel L Aldridge, Conceptualization, Data curation, Formal analysis, Investigation, Methodology, Writing – original draft, Writing – review and editing; Zachary Lanzar, Anthony T Phan, David A Christian, Ryan D Pardy, Data curation, Formal analysis, Investigation; Booki Min, Ross Kedl, Conceptualization, Resources, Methodology, Writing – review and editing; Christopher A Hunter, Conceptualization, Supervision, Funding acquisition, Writing – original draft, Project administration, Writing – review and editing

## Author ORCIDs

Daniel L Aldridge https://orcid.org/0000-0003-2224-5159
Anthony T Phan https://orcid.org/0000-0003-2935-6070
Ryan D Pardy https://orcid.org/0000-0002-0024-8888
Booki Min https://orcid.org/0000-0002-2151-9413
Christopher A Hunter https://orcid.org/0000-0003-3092-1428

## Ethics

This study was performed in strict accordance with the recommendations in the Guide for the Care and Use of Laboratory Animals of the National Institutes of Health. All of the animals were handled according to approved Institutional Animal Care and Use Committee (IACUC) protocols (#805045) of the University of Pennsylvania.

Reviewer #1 (Public review): https://doi.org/10.7554/eLife.105876.3.sa1
Reviewer #2 (Public review): https://doi.org/10.7554/eLife.105876.3.sa2
Author response https://doi.org/10.7554/eLife.105876.3.sa3

# Additional files

## Supplementary files

MDAR checklist

Source data 1. These are the raw data points are all graphed figures.

## Data availability

All flow cytometric analysis is quantified and represented within the manuscript; source data files have been provided.

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
