## [Editor Report · eLife Assessment]

The article presents **important** findings describing the role of IL27 in maintaining HSCs at steady state, and in emergency haematopoiesis in response to T. goodii by limiting the inflammatory monocyte outcomes. The evidence provided are **solid** and support that IL27 acts at the level of HSCs and not downstream. This study will be of interest to immunologists and hematologists, as well as infectious disease researchers.

---

## [Referee Report · Reviewer #1 (Public review)]

In the manuscript, Aldridge and colleagues investigate the role of IL-27 in regulating hematopoiesis during *T. gondii* infection. Using loss-of-function approaches, reporter mice, and the generation of serial chimeric mice, they elegantly demonstrate that IL-27 induction plays a critical role in modulating bone marrow myelopoiesis and monocyte generation to the infection site. The study is well-designed, with clear experimental approaches that effectively address the mechanisms by which IL-27 regulates bone marrow myelopoiesis and prevents HSC exhaustion. I have two minor comments that could enhance the conceptual framework of this study:

(1) The authors indirectly show that IL-27R expression on HSPCs is necessary for regulating HSC proliferation and preventing exhaustion. However, given that they have access to IL-27RFlox mice, they could cross these with Fgd5Cre mice to specifically delete IL-27R on long-term HSCs. This would provide direct evidence for the role of IL-27 signaling in LTHSCs during infection.

(2) Since memory T and B cells often home to the bone marrow, it would be interesting to consider the potential cross-talk between these cells, HSPCs, and IL-27 signaling during secondary *T. gondii* infection. A brief discussion of this possibility would strengthen the study's broader implications.

---

## [Referee Report · Reviewer #2 (Public review)]

Aldridge et al. demonstrate the important role of IL-27 in limiting emergency myelopoiesis in response to *Toxoplasma gondii* infection. Interestingly, IL-27 acts specifically at the level of early haematopoietic progenitors, inducing STAT signalling, which, in this case, dampens proliferation and preserves HSC fitness.

They used different mouse genetic models such as HSC lineage tracing, IL27 and IL27R-deficient mice to show that :

HSCs actively participate in emergency myelopoiesis during *Toxoplasma gondii* infection.

The absence of IL27 and IL27R increases monocyte progenitors and monocytes, mainly inflammatory monocytes CCR2hi.

At steady state, loss of IL27 impairs HSC fitness as competitive transplantation shows long-term engraftment deficiency of IL27 BM cells. This impairment is exacerbated after infection.

IL27 is produced by various BM and other tissue cells at steady state and its expression increases with infection, mainly by increasing the number of monocytes producing it.

This article highlights a new mechanism that acts directly at the level of early hematopoietic cells to limit over-inflammation during infection.

---

## [Author Response]

The following is the authors’ response to the original reviews.

**Reviewer #1 (Public review):**
In the manuscript, Aldridge and colleagues investigate the role of IL-27 in regulating hematopoiesis during *T. gondii* infection. Using loss-of-function approaches, reporter mice, and the generation of serial chimeric mice, they elegantly demonstrate that IL-27 induction plays a critical role in modulating bone marrow myelopoiesis and monocyte generation to the infection site. The study is well-designed, with clear experimental approaches that effectively adddress the mechanisms by which IL-27 regulates bone marrow myelopoiesis and prevents HSC exhaustion.
**Reviewer #2 (Public review):**
Summary:Aldridge et al. aim to demonstrate the role of IL27 in limiting emergency myelopoiesis in response to *Toxoplasma gondii* infection by acting directly at the level of early haematopoietic progenitors.They used different mouse genetic models, such as HSC lineage tracing, IL27 and IL27R-deficient mice, to show that:(1) HSCs actively participate in emergency myelopoiesis during *Toxoplasma gondii* infection.(2) The absence of IL27 and IL27R increases monocyte progenitors and monocytes, mainly inflammatory monocytes CCR2hi.(3) At steady state, loss of IL27 impairs HSC fitness as competitive transplantation shows long-term engraftment deficiency of IL27 BM cells. This impairment is exacerbated after infection.(4) IL27 is produced by various BM and other tissue cells at steady state, and its expression increases with infection, mainly by increasing the number of monocytes producing it.

Although it is indisputable that IL27 has a role in emergency myelopoiesis by limiting the number of proinflammatory monocytes in response to infection, the authors' claim that it acts only on HSCs and not on more committed progenitors (CMP, GMP, MP) is not supported by the quality of the data presented here, as described below in the weakness section. In addition, this study highlights a role for IL27 during infection, but does not focus on trained immunity, which is the focus of the targeted elife issue.

We thank the reviewer for these comments. We did try (and perhaps failed) to highlight that all cells within the HSPC category, which includes HSCs and MPPs, have the potential to contribute. The lack of IRGM1-RFP reporter expression in CMPs (Supp Fig5C) suggests that only HSCs and MPPs are progenitors that respond to IL-27 within the bone marrow, and thus that IL-27 signaling on these contributes to the effects observed on monopoiesis and peripheral monocyte populations. We have emphasized this in the revised manuscript, particularly in the introduction (line 82) and discussion (lines 469-472). While this manuscript does not focus solely on trained immunity, the impacts of infection regulating HSC differentiation and having a long-term impact on this compartment are a central theme of trained immunity. For example, Figure 6 and the supporting supplemental figures almost exclusively focus on the differentiation potential that is programed into LTHSCs by infection and the role of IL-27 in regulating this programing. Additionally, Figure 7 shows the long-term consequences of such training. The introduction and discussion have been modified to emphasize these connections to trained immunity.

Weakness(1) In Figure 4, MFI quantification is required. This figure also shows the expression level (FACS and RNA) in progenitors (GMP and CMP, GP, MP), which is quite similar to that of HSC at this level, so it is really surprising that CMP does not respond at all to IL27 (S5C).

As requested, we have included the MFIs, calculated as a fold change over control FMOs, in the revised manuscript. While HSPCs and CMPs show relatively similar RNA expression of Il27ra (Supp. Fig. 5 A), the levels of surface IL-27R expression by CMPs is lower than HSPCs (Fig. 4C, revised). Additional downstream progenitors (including GMPs) show highly reduced RNA expression and a corresponding low expression of the receptor protein. This is now more apparent with the quantified MFIs (Fig 4-5).

(2) Total BM was used to test the direct effect of IL27 on HSC. There could be an indirect effect from other more mature BM cells, even if they show lower receptor expression than HSC. This should be done on a different sorted population to prove the direct effect of IL27 on HSC. The authors need to look more closely at some stat-dependent genes or stat itself in different sorted cell populations, not just irgm1. It is also known that Stat is associated with increased HSC proliferation in response to IFN, which is the opposite of what is observed here.

We thank the reviewer for this question. We have found that the methanol fixation required to detect pSTAT disrupted the ability to stain for HSPCs by flow cytometry. Thus, we used the IRGM1 reporter, which we have found to be a sensitive and high-fidelity reporter of STAT1 activity while preserving epitope markers of HSPCs.

We agree that the use of bulk bone marrow in the in vitro stimulations could allow for the activation of non-HSPC cell types that are IL-27R+. This is now emphasized in the text. However, there are advantages to this bulk approach as it allows simultaneous analysis of all HSPC populations and downstream progenitors in the same cultures, allowing the ability to assess how the small numbers of IL-27R expressing lymphocytes present in these cultures respond (data that are now included, Supp. Fig. 5C). These cultures also allow a direct comparison of our IL-27R expression analysis with responsiveness to IL-27. Only a selection of the populations analyzed are shown in these data; however, all populations in Figure 4A were also analyzed in Supp. Fig. 5C. These data sets directly correlate receptor expression with sensitivity to IL-27. If this effect was indirect (i.e the ability of IL-27 to induce IFN-γ) then we would expect more robust expression of the IRGM1 reporter across other cell populations. However, while IFN-γ stimulates broad expression of IRGM1, the effects of IL-27 are restricted to HSPC and mature lymphocytes (Supp. Fig. 5C). In other words, the cells that express the highest levels of the IL-27R are most responsive to IL-27.

While we do not directly measure HSPC proliferation in these cultures, we agree with the reviewer that the decreased proportions of proliferating HSPCs seen in the absence of IL-27 during infection (Fig. 7A) is a complex data set. The reviewer is also correct that interferons can promote HSC proliferations; however, they can also promote cell stress, DNA damage, and even cell death of HSCs during chronic exposure (reviewed extensively in Demerdash, Y., et al. Exp Hematol. 2021. PMID: 33571568). Thus IFNs, much like IL-27, appear to regulate HSPCs with contextual importance, inducing their proliferation but also death. The activation of STAT1 and STAT3 by IL-27 may be at the core of some of these effects observed in our data, and we point out that IL-10, another activator of STAT1+3, has been shown to limit HSC responses to inflammation (lined 58-62), but we have also presented other possibilities in the discussion.

(3) The decrease in HSC fitness in IL27R KO at steady state could be an indirect effect of the increase in proinflammatory monocytes contributing to high levels of inflammatory cytokines in the BM and thus chronic HSC activation that is enhanced in response to infection. What is the pro-Inflammatory cytokine profile of the BM of IL27 OR IL27R deficient mice and of mixed chimera mice.

We thank the reviewer for this insightful comment. This was part of our stated rationale in generating the mixed WT:IL-27R-/- BM chimeras presented in Figure 2. In this mixed setting, there remained differences between the ability of the IL-27R sufficient and deficient stem cells to generate inflammatory macrophages. These results suggest that differences in the inflammatory environment do not account for the differences observed. This conclusion is further supported by the observation that the infection-induced levels of IFN-γ in the bone marrow are equivalent in the presence or absence of IL-27 (now included in the revised manuscript, Supp. Fig. 1F).

(4) Furthermore, the FACS profile of KI67/brdu of Figure 7 is doubtful, as it is shown in different literature that KSL are not predominantly quiescent as shown here, but about 50% are KI67-. This is also inconsistent with the increase of HSC observed in Figure 1. Quantification of total BruDU+ HSC and other progenitors is also important to quantify all cells that have proliferated during infection. As the repopulation of IL27-deficient BM is also lower in the absence of infection the proliation of HSC in IL27R KO mice in the absence of infection is also important.

The comment indicates that the reviewer is concerned that our staining for Ki67 is on the low end of reported literature (~10-50% of LSKs, depending on age of the mice and simulation (Thapa R, et al. Stem Cell Res Ther. 2023. PMID: 37280691; Nies KPH, et al. Cytometry A. 2018. PMID: 30176186)). Our stains were performed on cells from infected mice, which does alter the classic markers used to identify HSPCs. For this reason, we are stringent with our gating strategy and may be excluding more HSPCs than are included in other reports. We have included our FMO control in the revised manuscript to indicate our gating approach (Supp. Fig. 9A). While the population of Ki67+ HSPCs is low, these results were consistent between our experiments and provide data sets that are interpretable.

(5) The immunofluorescence in Figure 3 shows a high level of background and it is difficult to see the GFP and tomato positive cells. In this sense, the number of HSCs quantified as Procr+ (more than 8000 on a single BM section) is inconsistent with the total number of HSCs that a BM can contain (i.e., around 6000 per BM as quantified in Figure 1).

We agree with the reviewer and have found that there is a high level of background in these stains. We have thresholded these images, as described in our methods, to minimize this. Additionally, the increased numbers of Procr+ cells in the imaging vs our flow data is expected, and has been reported by others (Steinert, EM, et al. Cell. 2015. PMID: 25957682).

(6) The addition of arrows to the figure will help to visualise positive cells. It is also not clear why the author normalised the GFP+ cells to the tomato+ cells in Figure 3D.

We thank the reviewer for this comment and have added the suggested arrows. We have also included a more detailed explanation for our normalization strategy.

(7) Furthermore, even if monocytes represent a high proportion of IL27-producing cells, they are only 50% of the cells at 5dpi, as shown in Figure 3 and S4. Without other monocyte markers, line 307 is incorrect.

We thank the reviewer for this clarification and have adjusted the text accordingly.

(8) How do the authors explain that in Figure 1, 5-10% of labelled precursors and monocytes can give 100% of monocytes? This would mean that only labelled HSC can differentiate into PEC monocytes. 5

We thank the reviewer for their interest in this result. Monocytes and macrophages are some

**Reviewer #1 (Recommendations for the authors):**
I have two minor comments that could enhance the conceptual framework of this study:(1) The authors indirectly show that IL-27R expression on HSPCs is necessary for regulating HSC proliferation and preventing exhaustion. However, given that they have access to IL-27RFlox mice, they could cross these with Fgd5Cre mice to specifically delete IL-27R on long-term HSCs. This would provide direct evidence for the role of IL-27 signaling in LTHSCs during infection.

We appreciate this comment and did attempt this experiment with several HSPC specific Cres, including the Procr-cre (used elsewhere in the manuscript) and the MDS1-cre-ERT2 (Jackson Laboratory Strain #:032863). Unfortunately, validation revealed that deletion efficiency of the IL-27R with these HSCspecific Cre lines was inefficient, and so experiments are ongoing to enhance efficiency of the deletion and test alternative Cre lines (such as the Fgd5-cre).

(2) Since memory T and B cells often home to the bone marrow, it would be interesting to consider the potential cross-talk between these cells, HSPCs, and IL-27 signaling during secondary *T. gondii* infection. A brief discussion of this possibility would strengthen the study's broader implications.

We thank the reviewer for this opportunity. We have previously investigated the interplay between immune cells in the bone marrow (Glatman Zaretsky A, et al. Cell Rep. 2017. PMID: 28228257) and now include these possibilities in the discussion (line 465-470).

**Reviewer #2 (Recommendations for the authors):**
Minor points:(1) Figures 6F and 7B: should be shown as % of donor and not total number to clarify the lineage potency of LTHSC. The fact that the results of transplantation are separated into different figures makes it not easy to follow. To see if the increase in monocyte production by IL27 KO BM is specific, the percent of donorderived cells for other populations, such as lymphoid, but also in MP, and inflammatory monocytes, is necessary to confirm Figure 2.

Perhaps there has been a misunderstanding? In these plots, we are not analyzing mixed chimeras but single transfer chimeras into lethally irradiated hosts. Thus, the % of donor reaches ~80- 90%. However, to measure the actual output of the HSPCs, the cell number was necessary to compare amongst groups. Additional description is provided in the figure legends and in the text of the manuscript (lines 391-392, 434-436, 651-653, and 680-682).

(2) The heavy UMAP description is unnecessary. Responses As requested, we have reduced this description of how the UMAPs were derived.

As requested, we have reduced this description of how the UMAPs were derived